# Degenerative Changes in the Claustrum and Endopiriform Nucleus after Early-Life Status Epilepticus in Rats

**DOI:** 10.3390/ijms25021296

**Published:** 2024-01-20

**Authors:** Rastislav Druga, Pavel Mares, Martin Salaj, Hana Kubova

**Affiliations:** 1Institute of Anatomy, 2nd Medical Faculty, Charles University, 15006 Prague, Czech Republic; martin.salaj@lfmotol.cuni.cz; 2Laboratory of Developmental Epileptology, Institute of Physiology, Czech Academy of Sciences, 14200 Prague, Czech Republic; maresp@biomed.cas.cz; 3Institute of Anatomy, 1st Medical Faculty, Charles University, 12000 Prague, Czech Republic

**Keywords:** status epilepticus, claustrum, endopiriform nucleus, claustroamygdaloid complex, neurodegeneration, ontogeny

## Abstract

The aim of the present study was to analyze the location of degenerating neurons in the dorsal (insular) claustrum (DCL, VCL) and the dorsal, intermediate and ventral endopiriform nucleus (DEn, IEn, VEn) in rat pups following lithium–pilocarpine status epilepticus (SE) induced at postnatal days [P]12, 15, 18, 21 and 25. The presence of Fluoro-Jade B-positive neurons was evaluated at 4, 12, 24, 48 h and 1 week later. A small number of degenerated neurons was observed in the CL, as well as in the DEn at P12 and P15. The number of degenerated neurons was increased in the CL as well as in the DEn at P18 and above and was highest at longer survival intervals. The CL at P15 and 18 contained a small or moderate number of degenerated neurons mainly close to the medial and dorsal margins also designated as DCl (“shell”) while isolated degenerated neurons were distributed in the VCl (“core”). In P21 and 25, a larger number of degenerated neurons occurred in both subdivisions of the dorsal claustrum. The majority of degenerated neurons in the endopiriform nucleus were found in the intermediate and caudal third of the DEn. A small number of degenerated neurons was dispersed in the whole extent of the DEn with prevalence to its medial margin. Our results indicate that degenerated neurons in the claustrum CL and endopiriform nucleus are distributed mainly in subdivisions originating from the ventral pallium; their distribution correlates with chemoarchitectonics of both nuclei and with their intrinsic and extrinsic connections.

## 1. Introduction

Status epilepticus (SE) is often used experimentally to trigger epileptogenesis and the development of complex structural and functional changes resembling human temporal lobe epilepsy [1]. Both limbic (amygdala, claustrum, endopiriform nucleus, piriform cortex, entorhinal cortex, hippocampal formation) but also extralimbic structures undergo substantial neuronal loss and structural reorganization after SE in adult as well as in young immature animals [2,3,4,5,6,7,8,9].

The claustrum is a subcortical telencephalic structure present in all mammals examined—from insectivora to primates and humans [10]. Two principal parts of this nucleus can be distinguished in all mammals, namely the dorsal (insular) claustrum, which underlies the insular cortex and the ventral claustrum (piriform claustrum, endopiriform nucleus), which adjoins the piriform cortex [11,12,13,14,15]. Expression patterns of the developmental regulatory genes indicate that the claustrum, the endopiriform nucleus and a part of the amygdala comprise an entity called the claustroamygdaloid complex [16,17] and that derivatives of the ventral and lateral pallium can be distinguished in the claustroamygdaloid complex. A major part of the dorsal claustrum (the dorsolateral claustrum, claustrum proper), the basolateral amygdala, posterolateral cortical amygdalar area and dorsal part of the piriform cortex are considered derivatives of the lateral pallium, while the ventromedial claustrum (smaller, medial part of the dorsal claustrum adjoining the external capsule (see [17]), the endopiriform nucleus, several amygdalar nuclei and the ventral part of the piriform cortex are considered derivatives of the ventral pallium [16,17].

The claustrum nuclei (dorsal claustrum, endopiriform nucleus) contain mostly glutamatergic neurons and several subpopulations of GABAergic interneurons. Subpopulations of neurons within the CL and the ventral claustrum in rodents express GABA, calcium-binding proteins, neuropeptides and nitric oxide synthase (NOS) [12,15,18,19,20,21,22,23]. Calcium-binding proteins and neuropeptides in the CL and DEn are frequently colocalized [21,24]. Using single-cell RNA sequencing, it was demonstrated that claustrum contains two excitatory (glutamatergic) neuron subtypes, which differ in the expression of genes and form a core–shell organization [25].

In higher mammals, two principal parts of the claustrum (dorsal or insular and ventral or piriform claustrum) are distinguished [11,14], but in rodents, the traditional dorsal claustrum (CL) was recently parcelled into two subdivisions: the dorsal claustrum (DCl) and the ventral claustrum (VCl) [26]. These two parts differ in the expression of cadherins, calcium-binding proteins (parvalbumin, calretinin), the glutamate transporter GLUT2 and NOS. In contrast, the expression of cadherins might indicate three subdivisions [18,23,27]. Rat and guinea pig claustrum have been found to have complementary patterns of PV and CR immunoreactivity. The CL of the rat consists of two regions, the first being a CR negative zone in the core of the structure. This region is dorsally, medially and basally surrounded by CR-positive neuropil containing a small number of CR-positive neurons (shell). CR-negative zone corresponds to a strong PV-positive area containing a high density of PV-positive neurons and neuropil [4,12,23].

Similarly, the classical ventral (piriform) claustrum (the endopiriform nucleus) was further divided into the dorsal endopiriform nucleus (DEn), the intermediate endopiriform nucleus (IEn) and the ventral endopiriform nucleus (VEn) [26]. A recent analysis of the distribution of calcium-binding proteins and latexin in the CL and DEn of the short-tailed fruit bat proposes further subdivision of the shell subregion into four sectors [28].

Proteomic analyses indicate that the CL in the rat has a shorter anteroposterior extent and that the claustrum in rodents and primates is surrounded by layer VI of the insular cortex. Thus, the concept of the claustrum as an intracortically located structure (within layer VI), as originally introduced by Narkiewicz and Mamos [29], is supported and characterized by the expression of a specific protein [30].

Hodological analysis demonstrates that the CL has reciprocal connections with many neocortical regions, while the DEn has bidirectional connections with the piriform cortex and other limbic structures [13,15,31,32].

The functions of the CL were not yet sufficiently explained. Rich and bidirectional connections with many neocortical areas indicate that it may influence the responsiveness of these areas [33,34]. The claustrum might also integrate sensory information from many cortical areas and form a background for responses to complex stimuli [35].

Convergence of information from different levels of the olfactory system and from the amygdala occurs in the DEn. In addition, electrical and optical recordings indicate that olfactory and gustatory activity converges onto single neurons of the DEn [36,37,38].

The claustrum and piriform cortex play a role in temporal lobe epileptogenesis. The deep piriform region including the IEn, DEn and CL was identified as a region with an especially low threshold for the generation of epileptiform discharges. Microelectrode mapping indicates that the dorsal edge of the DEn is the site where these discharges are initiated [39]. In addition, the piriform cortex, IEn, DEn and CL are among the structures that exhibit severe pathologic changes in various epilepsy models in adult animals [2,3,8,9,40,41].

Our previous results demonstrated that the piriform cortex and the DEn and IEn are substantially damaged in young rats (P25) that survive lithium–pilocarpine SE, whereas the CL is relatively preserved [4]. More detailed data about the topography of degenerated neurons and about the time- and age-dependent progression of neuronal degeneration in the ventral and dorsal claustrum (DEn and CL, respectively) are lacking.

In the present study, we stained neurons in these structures undergoing degeneration using FluoroJade-B (FJB) an efficient fluorochrome [42]. This dye was used to identify not only the distribution of neuronal damage but also the timing of damage in the dorsal (insular) claustrum (DCl, VCl) and the ventral (piriform) claustrum (DEn, IEN, VEn) of immature rats after lithium–pilocarpine induced SE. An additional aim was to relate the distribution of degenerating neurons to the recently introduced parcellation of the dorsal claustrum [16,18,23,26,27] (Figure 1).

## 2. Results

Status epilepticus was induced in all age groups used. Individual groups are signed according to the age of SE induction. Mortality increased with age at SE. No mortality was observed in the P12 group, whereas the highest mortality (35 and 32%, respectively) was observed in P21 and P25 rats.

### 2.1. Distribution of Degenerating Neurons and Severity of Damage

No FJB-positive neurons were observed in the control animals regardless of age and interval.

#### 2.1.1. Severity of Damage and Distribution of Degenerating Neurons in the Dorsal (Insular) Claustrum (CL) in SE Animals

Both subdivisions of the CL differ significantly in the density of degenerating neurons. The majority of FJB-positive neurons were detected in the DCl (shell), whereas damage to the VCl (core) was rather negligible (Figure 2 and Figure 3). In the DCl the majority of survival intervals exhibited the largest density of degenerating neurons in P21 and/or P25 animals.

Two-way ANOVA revealed a significant effect of age at SE (F (4, 50) = 51.73; *p* < 0.0001) interval after SE (F (4, 50) = 35.25; *p* < 0.0001) and their interaction (F (16, 50) = 8.882; *p* < 0.0001) on the severity of damage in the CLD. The density of FJB-positive neurons (number of FJB-positive cells/mm^2^) increased with age at SE and peaked at the 24 h interval. FJB-positive neurons were not observed in P12 in any interval after SE. In P15 rats, only sparse FJB-positive neurons (<10 per anatomic area) were found in intervals of 12–48 h. In P18 rats, the density of FJB-positive neurons was significantly lower 24 h after SE compared to P21 (Figure 4A, left panel). The mean number of FJB-positive cells per anatomic area (area of the CLD) in P21 and P25 was 48 and 55, respectively.

In the VCl, (core) density of FJB-positive neurons was negligible or negative with the exception of P18, P21 and P25 animals 24 h after SE. In this interval, 3 to 14 FJB-positive neurons per anatomic area were detected. In other age- and interval groups FJB-positive neurons occurred only sporadically and prevailed in marginal parts of the VCl (Figure 3 and Figure 4A, right panel).

The area of neither the DCL nor VCL differed across individual age and interval groups (Figure 5A).

#### 2.1.2. Severity of Damage and Distribution of Degenerating Neurons in the Endopiriform Nucleus

FJB-positive neurons were detected in all age groups and at all intervals after SE. Two-way ANOVA revealed a significant effect of age at SE (F (4, 50) = 43.26; *p* = 0.0283), interval after SE (F (4, 50) = 2.965; *p* < 0.0001) and their interaction (F (16, 50) = 2.776; *p* = 0.0030) on the severity of damage expressed as a density of FJB-positive neurons (number of FJB-positive neurons per mm^2^). In all intervals, the density of positive cells was lower in the three youngest age groups compared to P21 and P25 animals. The lowest density of FJB+ cells was found in P12 and P15 rats. In P18, the density of labeled cells was higher compared to younger age groups, but still significantly lower than in P21 and P25 animals 24 h after SE. FJB-positive neurons were equally distributed along the whole rostrocaudal length of the DEN in the caudal part of the DEN (Figure 4 and Figure 6). In all intervals, FJB-positive neurons prevailed in the medial and basal parts of the DEN (Figure 3, Figure 4 and Figure 6).

Since P15 small number of degenerated neurons was dispersed in the IEN, while no FJB-positive neurons were detected in the VEN.

Two-way ANOVA revealed significant effects of age (F (4, 50) = 9.743; *p* < 0.0001), but not of interval after SE or age x and interval interaction on the DEN area. The DEN area tended to be higher in P21 animals in most of the intervals after SE, but post hoc analysis showed significant differences only between P21 animals and the two youngest groups of rats (Figure 5). The statistical differences have to be however interpreted with caution because they were observed only in some intervals after SE and data were obtained from a relatively small number of animals.

### 2.2. Characteristics of Degenerated Neurons

FJB-positive small, rounded and less frequently bipolar neurons were characteristic in all subdivisions of the claustrum (DCl, VCl, DEN, IEn) at short survival intervals. In contrast, neurons of various sizes (15–33 μm) and with triangular (pyramidal) and multipolar perikarya and a variety of somatodendritic patterns represented approximately 80% of FJB-positive cells within the DCl, VCl and DEN at longer intervals, especially in P18 and older animals (Figure 7).

At shorter intervals up to 24 h after SE, FJB-positive neurons exhibited intense staining of the cell body and proximal dendrites. At longer survival intervals (48 h and 1 week), the DCl and DEN contained a mixture of intensely and less intensely stained neuronal bodies. Some of the less intensely stained (paler) neurons were shrunken with fragmented processes. Additionally, there were dispersed small stained particles formed probably by disintegrated processes and axon terminals. These stained particles visible at a 1-week survival interval are responsible for a “dusty appearance” of the neuropil.

### 2.3. Distribution of Calretinin- and Parvalbumin-Positive Neurons

Immunostaining for calretinin (CR) and parvalbumin (PVA) demonstrates the complementary distribution of both calcium-binding proteins in the dorsal (insular) claustrum. The high density of PVA-positive neurons and neuropil was observed in the VCL (core subdivision). This subdivision is surrounded by the rim of CR-positive neurons in the DCL (shell subdivision). In this subdivision, which contained the majority of degenerating neurons, PVA-positive neurons were sparse (Figure 1). Our data are in line with previous studies.

Functioning at this age: The incidence and latency to the onset of continuous convulsions (i.e., SE) were registered. SE was interrupted after 1.5 h of continuous motor seizures by an intraperitoneal injection of paraldehyde (0.3 mL/kg in rat pups at P18 and younger, 0.6 mL/kg in animals at P21 and P25). After paraldehyde injection, the rats were subcutaneously injected with 0.9% NaCl (up to 3% of the body weight divided into 2–3 doses) to restore volume loss. For about 3–4 days after SE, animals 18 days old and older were fed a moist diet. The health status of animals was monitored daily until the end of the study.

Each age and interval group consisted of three animals. Control siblings (n = 2 per age and interval group) were treated with an equal volume of LiCl but the pilocarpine was replaced with saline. A corresponding dose of paraldehyde was administered 2 h after saline injection.

## 3. Discussion

LiCl/pilocarpine-induced status epilepticus (SE) leads to the development of spontaneous recurrent seizures, cognitive deficits and behavioral alterations and extensive brain damage. It is a widely accepted model of temporal lobe epilepsy. Temporal lobe epilepsy in humans is a complex disorder in which seizures start or involve one or both temporal lobe structures in the brain, specifically the hippocampal formation and amygdalar nuclei. In many patients, temporal lobe epilepsy is associated with a high prevalence of psychiatric comorbidities like cognitive impairment, depression and emotional disturbances. It has been hypothesized that both TLE and its psychiatric comorbidities share common neuropathological and neurobiological aspects. In animal models, several other structures functionally related to the hippocampus and amygdala like parahippocampal cortices, piriform cortex and claustral complex are also damaged. In addition to typical temporal lobe structures, distant nuclear complexes like thalamic nuclei and several neocortical areas hodologically related to hippocampal and amygdalar circuits are also damaged [1,5,43].

The present study provides evidence of region-specific neuronal damage in the claustrum. Neuronal degeneration in the CL is an age- as well as survival interval-dependent process affecting all age categories. Degenerated neurons were detected in both subdivisions of the CL (DCL, VCL) but significantly prevailed in the DCl as well as in the endopiriform nucleus (DEn, IEn) at various intervals after lithium–pilocarpine induced SE. The number of degenerated neurons in the CL considerably increased in older animals (P21 and P25). A small number of degenerated neurons was detected in the CL (DCl) already in P12 and P15 pups. In older animals (P21 and P25) the number of positive neurons increased in the DCl but also in the VCl. In all groups of animals, FJB-positive neurons within the dorsal claustrum shared a similar topography; that is, in younger animals, they prevailed in the DCl (shell), and in older pups, a small number of degenerated neurons disseminated also to the VCl (core). The central part of the VCl was in younger animals (P15, P18) almost devoid of FJB-positive neurons. This part of the VCl contains many parvalbumin-immunoreactive neurons and a patch of strongly positive parvalbumin-immunoreactive fibers and terminals [19,23]. Very low immunostaining for calretinin is characteristic of the same area of the VCl, even though a small number of calretinin-immunoreactive neurons was detected in the periphery of this region (see Figure 2 and Figure 3). This pale focal area (core) is devoid of calretinin-immunoreactive fibers and puncta and is surrounded medially and laterally by a rim of stronger calretinin-immunoreactive neuropil in the rat as well as in the mouse [12,23]. In addition to an almost complementary distribution pattern of parvalbumin and calretinin within the central area (core) of the CL corresponding to the VCl [26], this part of the CL is characterized by strong cadherin in older animals (P21, P25) (Cad8, rat), whereas there is little neuronal NOS and vesicular glutamate transporter VGLUT2 [16,27,44]. The differences in neuronal damage between shell and core subdivision of the CL in younger and older animals may be related to different structures and vulnerability of local neuronal circuits [45]. It should be taken into consideration that claustro-cortical projecting neurons within DCl (shell subdivision) and VCl (core subdivision) in mice differ in their gene expression and cortical targets. It has been shown that neurons projecting to the retrosplenial cortex are located in the core subdivision of the insular claustrum, while neurons projecting to the lateral entorhinal cortex were found in the shell subdivision [25]. In our experiments, the core subdivision of the claustrum was in younger animals almost preserved while the majority of degenerated neurons were found in the shell subdivision. Such distribution of degenerated neurons within the insular claustrum indicates that neurons projecting to the limbic structures are in younger pups more vulnerable to SE.

The differences in the distribution of degenerated neurons in the subdivisions of the CL (VCl, DCl) and in the DEn after SE may be associated with specific hodological, neurochemical and developmental features of both nuclei.

The neuronal damage in the DEn was heavier than that in the CL and differed significantly between age groups. A small number of FJB-positive neurons (with low densities) was characteristic for the P12 and P15 age groups. In older animals, the number of degenerated neurons increased and peaked at P21.

The DEn is reciprocally connected with the piriform cortex and several other cortical formations [31,37,46]. These projections are largely excitatory and might provide a substrate for regenerative feedback interactions. Epileptiform activities generated in the DEn can drive, via these massive projections, paroxysmal activity in the overlying piriform cortex and back to the DEn [31,37,39,46,47]. It is possible that hyperactivity and the synchronization of synaptic activity in these circuits lead to an increase in glutamate release with a subsequent cascade of neurotoxic events resulting in neuronal degeneration. The specific membrane properties of the DEn neurons may contribute to the susceptibility of this nucleus to epileptiform activity [48]. The other characteristics of the neuronal mechanism within the DEn that explain the susceptibility of the DEn to seizure induction and propagation and eventually to neuronal damage were recently revised [37].

The existence of long rostrally directed associative projections within the DEn which are supposed to be glutamatergic may also contribute to the synchronization of neuronal hyperactivity, glutamate neurotoxicity and consecutive neuronal degeneration in the whole anteroposterior extent of the DEn [46]. Such associative projections were never demonstrated within the insular claustrum. The prevalence of neuronal degeneration in the caudal two thirds of the DEn might be explained by the additional influence of excitatory projections from several amygdalar nuclei (amygdalohippocampal area and other cortical amygdalar nuclei). These projections terminate in the intermediate and caudal part of the DEn [38].

The present study failed to demonstrate some specific features of neuronal degeneration in the dorsal part of the DEn that might be related to its specific role in the initiation of epileptiform activity [49]. A higher density of degenerated neurons, indicating a higher level of excitotoxicity, was evident in older animals (P18 and older) not only in the dorsal part of the DEn, but also in the medial and basal part of the nucleus. Neuronal degeneration within the DEn displays characteristics of a rather chronic process because the DEn in P18 and older animals contained a moderate number of FJB-positive neurons even 1 week after SE. In contrast to DEn, the IEn contained only a small number of degenerated neurons in all age groups and survival intervals. Negative findings were evident within the VEn.

It appears that the distribution pattern of degenerating neurons within the CL as well as in the DEn also has developmental relations. Degenerated neurons in the CL prevailed in the medial part of the VCl and in the DCl, which are probably derivatives of the ventral pallial histogenetic division. The lateral part of the VCl (called also dorsolateral claustrum, Cld) which is almost free of degenerated neurons is considered by Medina et al. [16] to be a derivative of the lateral pallial histogenetic division of the embryonic telencephalon. According to this developmental concept, the DEn which exhibited massive neuronal degeneration in the majority of survival intervals in our experiments is also considered a possible derivative of the ventral pallium. Thus, it appears that degenerated neurons within the CL (DCl, medial margin of the VCl) and the DEn are distributed predominantly in derivatives of the ventral pallium.

Comparison of the distribution of degenerated neurons in the dorsal and ventral claustrum with expression of Nurr1 (orphan nuclear receptor) and latexin indicated that Nurr+/Latexin- neurons prevailed in the parts of the claustral complex containing in our experiment FJB-positive neurons (DEn, DCl) [50,51,52,53].

The expression of a recently introduced marker of the CL, Gng2, indicates that in the rat hemisphere, the CL is discernible only at striatal levels and is surrounded medially and laterally by layer 6 insular cortex cells [30]. The Gng2-rich area probably corresponds to the lateral part of the subdivision of the CL designated by [26] as VCl. A small number of degenerating neurons were observed in this part of the CL in our experiments. In contrast, the medial part of the VCl and DCl contained degenerated neurons in an age- and survival interval-dependent manner.

The dynamics of neuronal degeneration in the DEn was similar to that in the CL (DCl) but the number of degenerated neurons in the DEn exceeded those in the CL. Larger neuronal damage of the DEn may be related to several hodological, cytochemical, structural and functional features. Among them, the pattern of local inhibitory interneurons may represent an important factor influencing the neuronal degeneration process. The core subdivision of the CL (VCl) contains many parvalbumin-immunoreactive neurons, a plexus of PV-ir fibers and a focus of parvalbumin-immunoreactive terminals, while PV-ir neurons are less frequent in the DEn [19]. In addition, the preservation of neurons within the core subdivision may be influenced by the synaptic organization of the neuronal circuits of this subdivision. The relationship among claustro-cortical neurons and PV-positive inhibitory neurons and feedforward inhibition of projecting neurons may represent substrate, which could contribute to the preservation of the core subdivision [45].

The DEn contains a large number of neuropeptide Y-positive neurons and the dorsal-most part of the DEn contains a large number of calretinin-positive boutons.

Summary and methodological considerations:

Our study demonstrated that there are rare degenerating neurons in both parts of the claustrum (DCl, DEn) if SE was elicited at the age of 12 days. Their number substantially increased if SE was induced in 18-day-old and older rats. In all age groups and survival intervals, the dorsal endopiriform nucleus (DEn) exhibited a higher number of FJB-positive neurons than the dorsal claustrum (DCl, VCl) (Figure 5 and Figure 6).

Taken together, our findings confirm the higher resistance of the immature brain to SE-induced damage. Several animal studies have already confirmed an increase in neurodegeneration with the age at SE. In rodents two weeks old or younger, damage to the hippocampus, amygdala complex or thalamus is small or even minuscule and the extent of neurodegeneration as well as the number of damaged structures increases with age at SE induction. In 3-week-old or older rats is comparable to those seen in adults [54,55,56].

The duration of SE together with the treatment chosen for termination of SE critically affects the severity and pattern of damage. Clinical studies have clearly shown that delayed treatment of SE is associated with an increased risk of morbidity and mortality as well as with a risk of treatment failure [57]. Clinical experience are supported by animal experiments showing a direct link between the duration of SE, its sequelae and the capability of treatment to stop seizure activity [58,59]. However, it also has to be emphasized that certain medications commonly used to terminate SE have been found to aggravate neuronal damage in immature rats [60]. In our experiments, a single dose of paraldehyde was administered after 1.5 h of ongoing motor seizure activity. In used doses, paraldehyde does not induce neurodegeneration in naïve P12 rats. Given treatment suppressed motor convulsions but it does not completely stop seizure activity in EEG [61]. In this respect, the SE model used for this study represents the model of long-lasting refractory status epilepticus associated with more serious consequences.

## 4. Materials and Methods

### 4.1. Animals

Male Wistar albino rats at P12, P15, P18, P21 and P25 postnatal days (P0 defined as the day of birth) were used. Animals (n = 125) were maintained with their dams on a 12/12 h light/dark cycle under controlled temperature (22 ± 1 °C) and humidity (50–60%), with free access to food and water until the end of the experiment. The experiments were approved by the Animal Care and Use Committee of the Institute of Physiology to be in agreement with the Animal Protection Law of the Czech Republic, which is fully compatible with European Commission Council directives 86/609/EEC.

### 4.2. Induction of Status Epilepticus

SE was induced by pilocarpine hydrochloride (#P6503, Sigma-Alrich^®^ Brand, Merck KGaA, Darmstadt, Germany; 40 mg/kg i.p.) in 5 age groups of rats: 12- (P12), 15- (P15), 18- (P18), 21- (P21) and 25-days (P25) old (pretreated 24 h earlier with lithium chloride (#L9650, Sigma-Alrich^®^ Brand, Merck KGaA, Darmstadt, Germany; 3 mEq/kg i.p.). Animals were observed for at least 2 h after pilocarpine administration. During experiments with 12- and 15-day-old pups, the temperature in Plexiglas cages used for observation was maintained at 32 ± 2 °C using an electric heating pad connected to a digital thermometer to compensate for immature thermoregulatory functioning at this age. The incidence and latency to the onset of continuous convulsions (i.e., SE) were registered. SE was interrupted after 1.5 h of continuous motor seizures by an intraperitoneal injection of paraldehyde (#P5520, Sigma-Alrich^®^ Brand, Merck KGaA, Darmstadt, Germany; 0.3 mL/kg in rat pups at P18 and younger, 0.6 mL/kg in animals at P21 and P25). After paraldehyde injection, the rats were subcutaneously injected with 0.9% NaCl (up to 3% of the body weight divided into 2–3 doses) to restore volume loss. For about 3–4 days after SE, animals 18 days old and older were fed a moist diet. The health status of animals was monitored daily until the end of the study.

Each age and interval group consisted of three animals. Control siblings (n = 2 per age and interval group) were treated with an equal volume of LiCl but the pilocarpine was replaced with saline. A corresponding dose of paraldehyde was administered 2 h after saline injection.

### 4.3. Histology

Tissue preparation: Rats of all age groups were killed 4, 12, 24, 48 h and 1 week after SE. Rats were overdosed with 20% solution of urethane (#U2500, Sigma-Alrich^®^ Brand, Merck KGaA, Darmstadt, Germany; 2.5 g/kg, i.p.) and perfused with phosphate-buffered saline (PBS, pH 7.4), followed by 4% paraformaldehyde (#P6148, Sigma-Alrich^®^ Brand, Merck KGaA, Darmstadt, Germany; in 0.1 M phosphate buffer (pH 7.4, 4 °C). The brains were removed from the skull, post-fixed for 3 h and then cryoprotected in graded sucrose (10%, 20%, and 30% in PBS). The brains were frozen in dry ice and stored at −70 °C until cut. A series of 50 μm thick coronal sections were prepared for further processing.

FuoroJade B staining: To detect degenerating neurons, a 1-in-5 series of sections was mounted on gelatin-coated slides and processed for FJB histochemistry according to [41,42]. Sections were examined with an epifluorescence microscope using flourescein thiocyanate filter sets. To better delineate the cytoarchitectonic boundaries of the claustrum and adjoining cortical areas, parallel sections were stained with cresyl violet.

Immunohistochemistry: Adjacent sections were processed immunohistochemically with antibodies raised against parvalbumin (mouse monoclonal, dilution 1:10,000, #P3088, Sigma-Alrich^®^ Brand, Merck KGaA, Darmstadt, Germany), or calretinin (mouse monoclonal, 1:8000, #MAB1568, Merck, NJ, USA) using the avidin–biotin method described previously in detail [12]. As a positive control, a thalamic section from an adult rat that experienced SE 24 h earlier was included into each set of immunostainings.

### 4.4. Parcellation of the Claustrum

For the description of the distribution of the degenerated (FJB-positive) neurons we used the parcellation of the rat claustrum according to Paxinos and Watson [26]. According to this parcellation, the dorsal (insular) claustrum (CL) was further subdivided into claustrum dorsale (DCl) and claustrum ventrale (VCl). Tracings of the CL, DEn and adjoining structures from adjacent series of Nissl stained sections and sections immunostained for parvalbumin and calretinin were used for identification of VCL and VCD borders (Figure 1).

### 4.5. Semiquantitative Analysis

Only neurons emitting intense yellow–green fluorescence that distinctly exceeded the background of the sections were included in a semi-quantitative analysis of damage severity. FJB-positive cells were counted in the dorsal endopiriform nucleus (DEn) and in the CLD and CLV subdivisions of the claustrum at 20-fold magnification directly from the sections using a microscopic grid. Counting of FJB+ cells in DEn was performed at three anteroposterior levels corresponding with Paxinos and Watson (2007) (23) AP +1.8 to AP −4.0. At each level, FJB^+^ neurons were counted per the anatomic area in three to four sections. The size of each anatomic area was assessed using the Olympus BX51 microscope (Tokyo, Japan) and QuickPHOTOMicro 2.3 software (Promicra, Prague, Czech Republic). The cytoarchitectonic boundaries were verified using adjacent Nissl stained sections and the density of FJB+ cells (number of cells per mm^2^) was calculated.

Degenerated neurons in the dorsal claustrum were counted in the CLD and CLV separately at the level AP 1.8–0.3 [26] at three consecutive sections.

### 4.6. Statistics

At the beginning of this study, simple randomization was used to assign each animal in individual age groups to a particular treatment and interval group. Data acquisition and analysis were conducted blinded to the treatment. Data were analyzed using GraphPad Prism 8 (GraphPad Software, Boston, MA, USA) software. Two-way ANOVA was used to identify the main effect of SE. Whenever a significant interaction was identified, the data were subjected to Tukey’s post hoc test. *p*-value < 0.05 was required for significance.

## 5. Conclusions

Early-life status epilepticus leads to neurodegeneration in the claustral complex. The extent and distribution of degenerating, FJB-positive neurons is highly dependent on the age at SE induction and intervals after SE. The severity of damage increases with age at SE and peaks at 24 h after SE. In the dorsal (insular) claustrum degenerated neurons prevailed in the calretinin positive zone (DCl, i.e., subdivision shell). Low density or almost absence of FJB-positive neurons was observed in the VCl, (i.e., subdivision core) with a high density of parvalbumin-positive neurons suggesting its protective role against SE-induced damage.

## Figures and Tables

**Figure 1 ijms-25-01296-f001:**
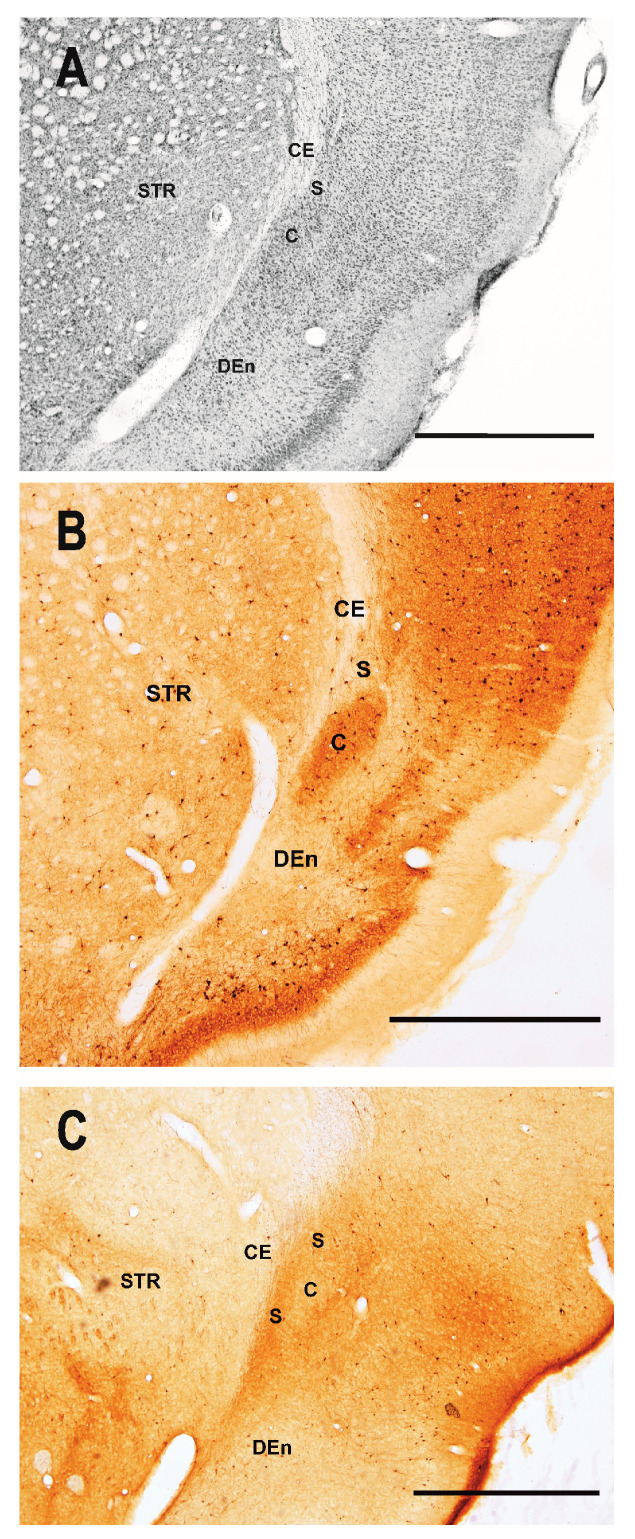
Low-power photomicrograph illustrating distribution of parvalbumin- and calretinin-positive neurons in the claustrum and dorsal endopiriform nucleus used for parcellation of the claustrum. (**A**) Shows cresyl-violet stained section indicating subdivisions of the claustrum. (**B**) Demonstrates the high density of parvalbumine (PVA)-immunopositive neurons in the ventral claustrum (i.e., in the “core”—C). In contrast, only sparse PVA-positive neurons were observed in the dorsal claustrum (i.e., in “shell”—S). The calretinin immunostaining is shown in (**C**). Abbreviations: CE—external capsule; C—dorsal claustrum, subdivision core; S—dorsal claustrum, subdivision shell; DEn—dorsal endopiriform nucleus, STR—striatum. Bar—500 μm.

**Figure 2 ijms-25-01296-f002:**
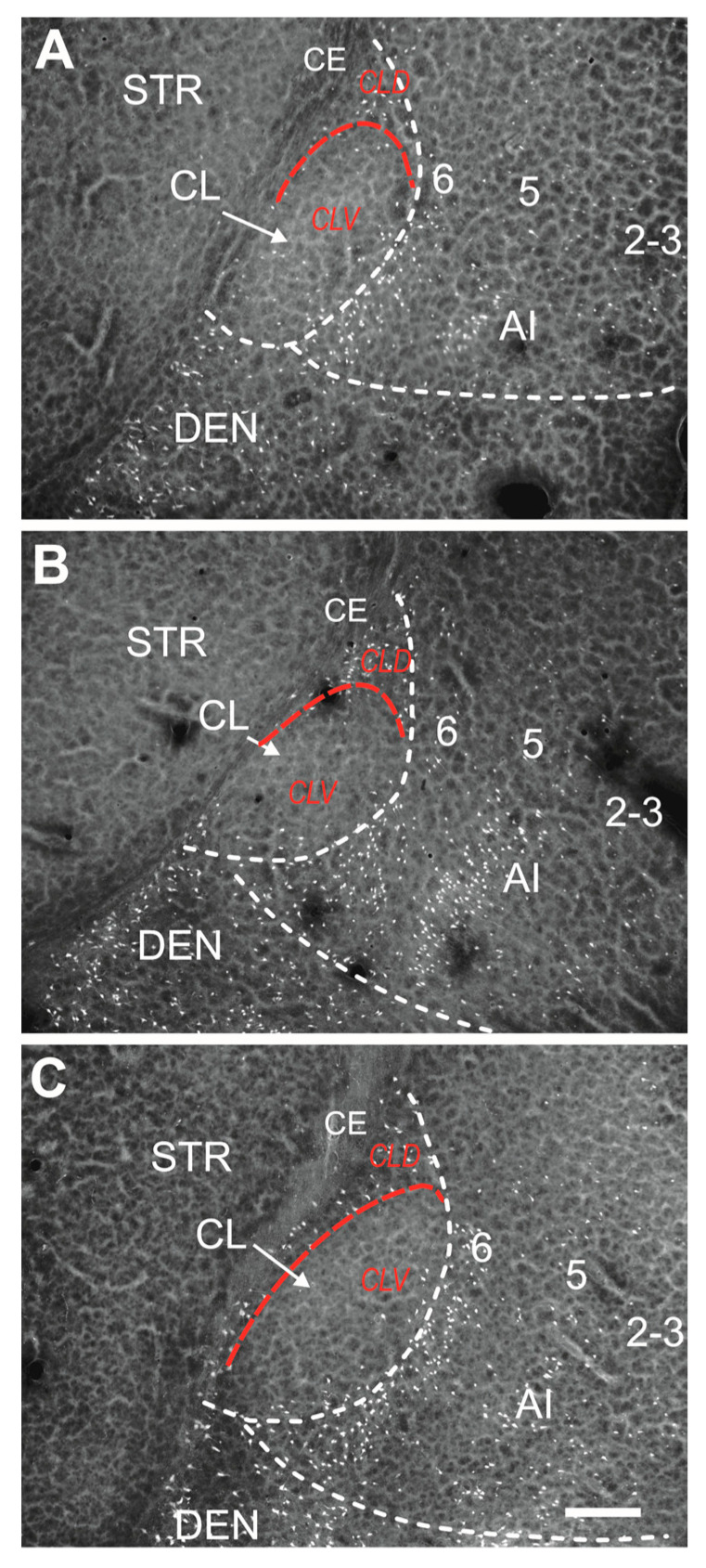
Distribution of FJB-positive neurons in the dorsal claustrum in the subdivisions shell (CLD) and core (CLV)(white dashed lines). Subdivisions are separated with dashed red lines. Panel (**A**) shows distribution and density of FJB-positive neurons in 18-day-old animal surviving 24 h after SE. Panel (**B**) illustrates neuronal damage in 21-day-old and Panel (**C**) in 25-day-old animals both in intervals 24 h after SE. Abbreviations: AI—agranular insular cortex, CE—external capsule, CLD—dorsal subdivision of the claustrum (shell), CLV—ventral subdivision of the claustrum (core), DEN—dorsal endopiriform nucleus, numbers indicate layers of the insular cortex. Bar—200 µm.

**Figure 3 ijms-25-01296-f003:**
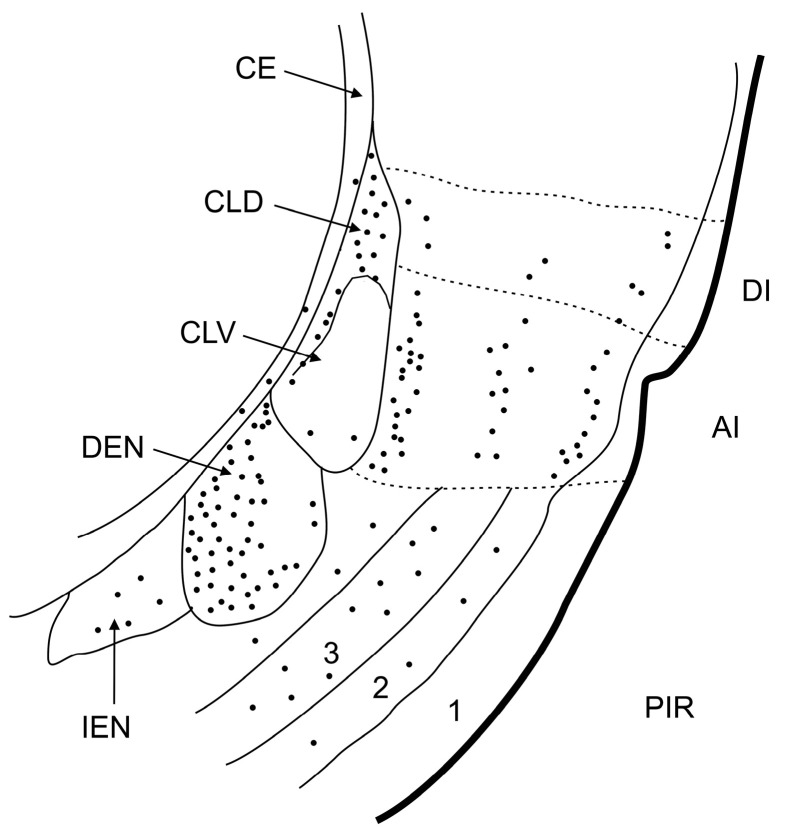
Schematic picture illustrating the parcellation of the claustrum and distribution of FJB-positive neurons (dots) in the claustral nuclei and adjoining cortical areas. Abbreviations: AI—agranular insular cortex, CE—external capsule, CLD—dorsal subdivision of the claustrum (shell), CLV—ventral subdivision of the claustrum (core), DEN—dorsal endopiriform nucleus, DI—disgranular insular cortex, IEN—intermediate endopiriform nucleus, PIR—piriform cortex, numbers indicate layers of the piriform cortex.

**Figure 4 ijms-25-01296-f004:**
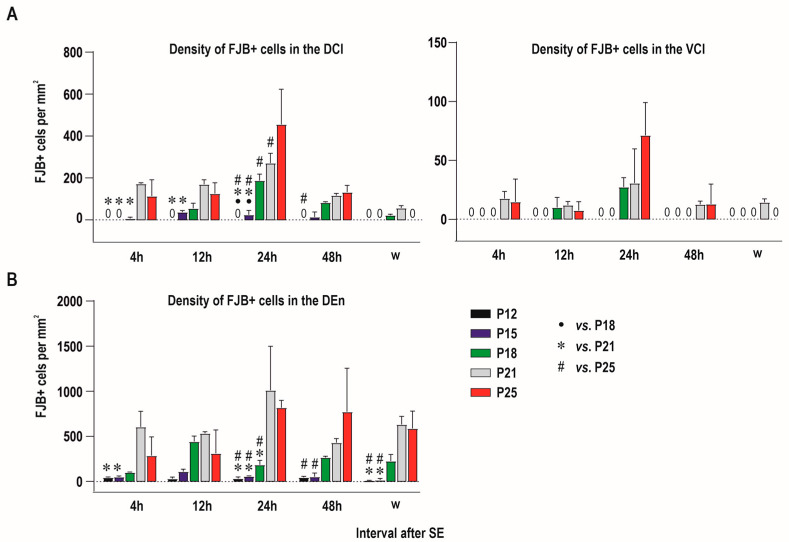
Graphs showing density (average number of FJB-positive neurons per mm^2^—abscisae) in individual age (see inset on the bottom part of the graph) and interval groups (intervals after SE—ordinatae) in the claustrum in both DCl and VCl subdivisions (**A**) and in the dorsal endopiriform nucleus (DEN) (**B**). Results are presented as mean ± SD.

**Figure 5 ijms-25-01296-f005:**
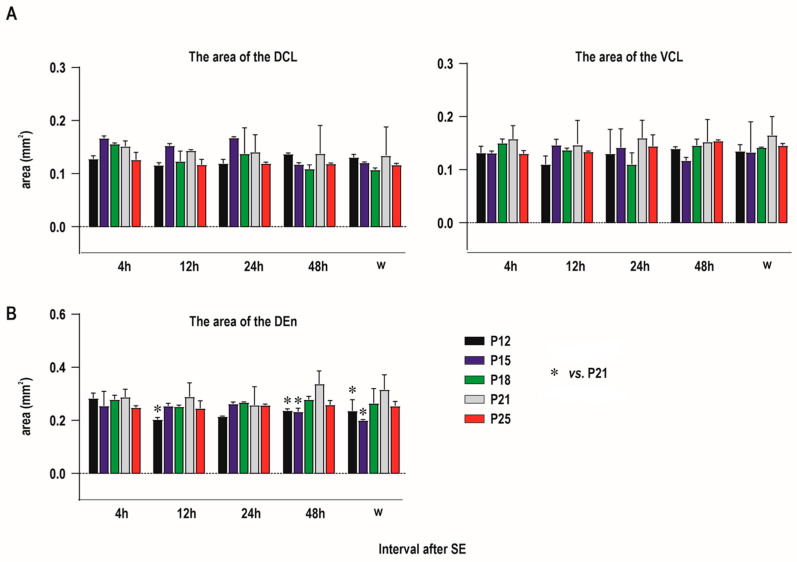
Graphs showing areas (in mm^2^—abscisae) of both the claustrum (both DCl and VCl subdivisions, (**A**)) and DEN (**B**) in individual age (see inset on the bottom part of the graph) and interval groups (intervals after SE—ordinatae). Results are presented as mean ± SD.

**Figure 6 ijms-25-01296-f006:**
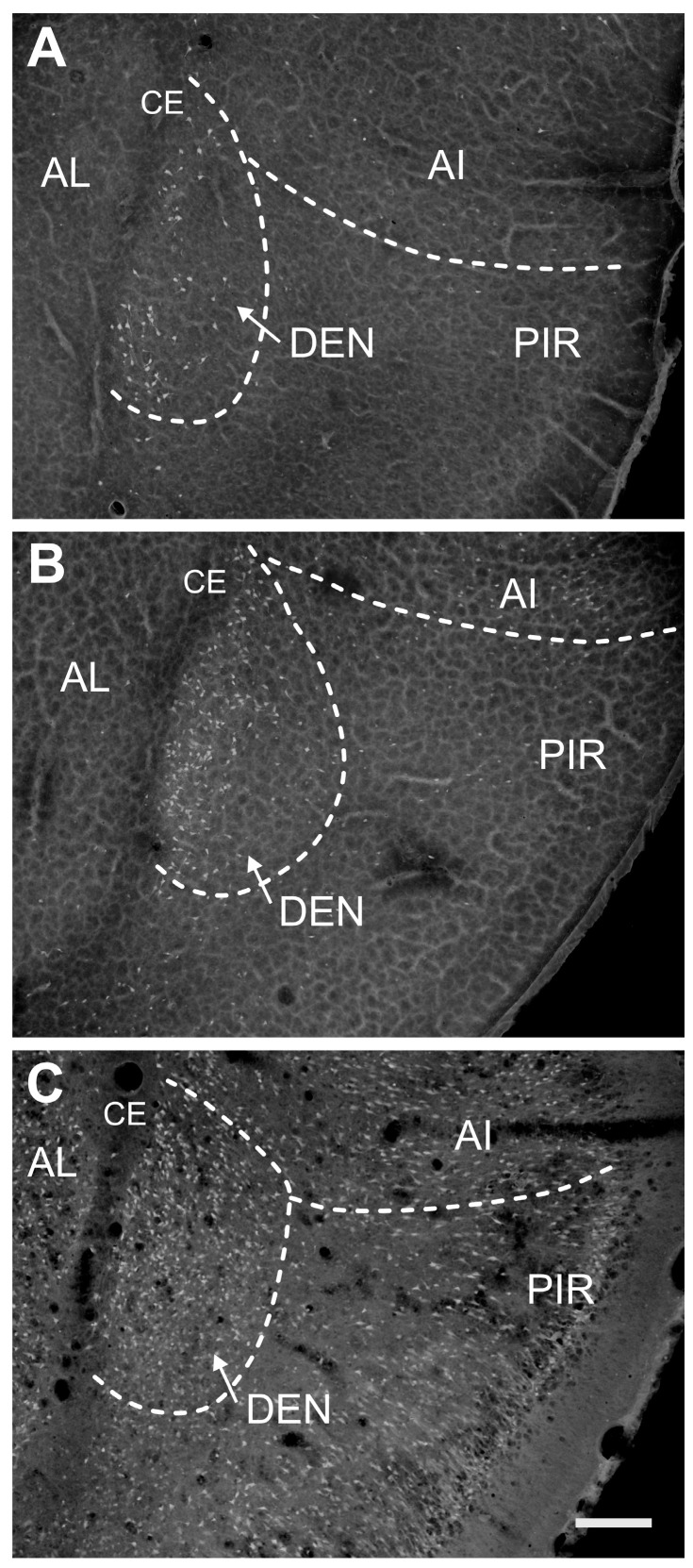
Distribution of FJB-positive neurons in the dorsal endopiriform nucleus (DEN) in 18-day-old (**A**), 21-day-old (**B**) and 25-day-old (**C**) rats 24 h after SE. All details as in Figure 2. Abbreviations: AI—agranular insular cortex, AL—lateral amygdalar nucleus, CE—external capsule, DEN—dorsal endopiriform nucleus, PIR—piriform cortex. White dash lines denote individual brain structures. Bar—200 µm.

**Figure 7 ijms-25-01296-f007:**
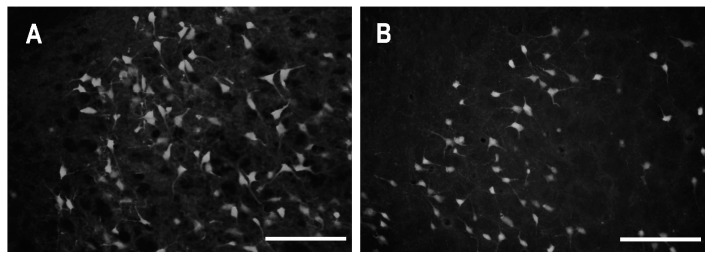
Somatodendritic morphology of FJB-positive neurons. (**A**) Morphology of degenerating neurons in the ventral claustrum, subdivision DEN in 25-day-old rats surviving 24 h after SE. (**B**) Morphology of degenerating neurons in the dorsal claustrum, subdivision DCl (shell) in the same animal. Bar 100 µm.

## Data Availability

All data and original materials are available in the Laboratory of Developmental Epileptology, IPHYS.

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
