# Peer review of "Degenerative Changes in the Claustrum and Endopiriform Nucleus after Early-Life Status Epilepticus in Rats"

_ijms, 2024, doi:10.3390/ijms25021296_

Round 1
Reviewer 1 Report
Comments and Suggestions for Authors
This is a review of manuscript ijms-2744770 “Degenerative changes in the claustrum and endopiriform nucleus after early-life status epilepticus in rats.” by Druga et al. The authors exhibited neuronal degenerative changes in the dorsal claustrum and the dorsal endopiriform nucleus in the pilocarpine-induced status epilepticus young rats.
This reviewer concerned critical points that the author showed only degenerative neuronal numbers and its characteristics.
The authors should exhibit data that what the difference between DCl or DEN and VCl is, and why FJB+ neurons were increased in DCl and DEN at P21 or P25 compared to at P18, etc.
The authors showed quantitative data per anatomic area, however, the size of anatomical area should depend on ages. They must show quantitative data per actual area (mm2).
SE severity score should be showed and discussed the relatives to severity of neuronal degeneration.
Author Response
Authors would like to thank for helping them to improve and clarify the manuscript.
- Several laboratories including ours already documented that the severity of morphological as well as functional sequalae of status epilepticus (SE) increases with age at SE onset. For long time, rodents younger than two weeks were considered to be resistant to seizure induced neuronal loss, behavioural alterations and epileptogenesis. Introduction and increased availability of new technologies (MRI, long-time EEG monitoring, new ways of the detection of neurodegeneration or DNA damage) allowed us to detect neuropathological and functional alterations even in animals exposed to SE shortly after birth. The first evidence showing the effect of age on a severity of damage as well as the clear proof of neuronal damage in two week old rats was published 1998 (Sankar R, Shin DH, Liu H, Mazarati A, Pereira de Vasconcelos A, Wasterlain CG. Patterns of status epilepticus-induced neuronal injury during development and long-term consequences. J Neurosci. 1998 Oct 15;18(20):8382-93.). Since then, the difference in sensitivity to seizure-induced damage between rodents younger than two week three weeks old and older was repeatedly demonstrated by several laboratories in various brain structures (the hippocampus, thalamus, etc.). To point out the effect of age and SE duration on the severity of neuronal damage we added the brief explanation (methodological consideration) into discussion part. We also pointed out the role of treatment used to stop SE in the severity of neuronal loss.
- According to your suggestion, we calculated the density of FJB+ neurons per mm2 and used this parameter instead of the total number of labelled cells for statistical evaluation (Results 2.2.1. and 2.2.2.; Figure 4). We also compared areas of individual parts among age and interval groups (Results 2.2.1. and 2.2.2.; Figure 5). This comparison revealed only small differences in the area of the DEN between two youngest age groups (P12 and P15) and P21. However, these results have to be interpreted with caution, because difference was observed only in some intervals after SE and data were obtained from relatively small number of animals. It has to be also emphasized that there was the same pattern of developmental differences in both the density of FJB+ neurons and their total number per anatomic area.

Reviewer 2 Report
Comments and Suggestions for Authors
The authors analyzed the location of degenerating neurons in the dorsal (insular) claustrum and in the dorsal, intermediate and ventral endopiriform nucleus in rat pups following lithium-pilocarpine status epilepticus (SE) induced at postnatal days. The study tried to improve our knowledge about the pathogenetic mechanisms of SE. The manuscript is clear and well-written: therefore, I have few comments for the authors:
Introduction: please, specify shortly the various role of the different anatomical structures involved in the SE.
Discussion: the authors could speculate possible link between their results and what could happen in human brain during SE: it could be of a certain interest for the readers. Moreover, the authors must underline the important and crucial role of time for the development of anatomical damages: the authors must read and cite the paper by Verrotti A et al. Nat Rev Neurol. 2018 May;14(5):256-258
Comments on the Quality of English LanguageThe language is good.
Author Response
Many thanks for your suggestions and recommendations that helps us to improve the manuscript.
- In the first paragraph of discussion, we briefly described major differences in distribution of brain lesions induced by SE in our model and brain lesions described in patients with TLE. In our model, epileptogenesis is triggered by relatively long-lasting (1.5hour), continual convusions, i.e. severe brain insult associated with extensive neuronal injury in the temporal as well as extratemporal structures. Apparently, the brain damage is more extensive compared to human TLE. Besides, severity of damage as well as the number of involved brain structures is highly dependent on the age at SE onset. Interestingly, behavioral outcome of SE in animal models of TLE are very similar to psychiatric comorbidities described in human patients. They involve cognitive deficits, emotional disturbances and anxiety suggesting that both epilepsy and its comorbidities are sharing the same neuropathological and neurobiological aspects.
- According to your suggestion we included one paragraph concerning the role of duration of SE in the severity of neuropathological alterations and response to treatment into discussion part (methodological considerations). We also added a note concerning the significance of age at SE onset in the severity of seizure-induced brain injury.
